# OPTIMAL WINDOWING OF MR IMAGES USING DEEP LEARNING: AN ENABLER FOR ENHANCED VISUALIZATION

**Deepthi S**                                                    deepthi.s@ge.com
**Dheeraj Kulkarni**                                  dheeraj.kulkarni@ge.com
**Jignesh Dholakia**                                   Jignesh.Dholakia@ge.com
*GE Healthcare*

## Abstract

Window width (WW) and window level (WL) adjustments aid in visualizing anatomies with a suitable contrast. However, the presence of background noise in MR images biases the calculation of default WW/WL values since it necessitates a trade-off between enhancing contrast of foreground/anatomy of interest vs suppressing background/ outside the anatomy of interest. This paper proposes an intelligent algorithm to improve the automatic computation of WW/WL and provide better control for user defined windowing. This is achieved by first eliminating the background pixels using a Deep Neural network and then computing WW/WL.

**Keywords:** Window width, window level, deep learning, intelligent windowing, background suppression, MRI, U-net

## 1. Introduction

Windowing computations are imperative for visualizing anatomies with a suitable contrast. This involves the determination of two preset values called window width (WW) and window level (WL) and using them to transform the intensity distribution of the image. However, the presence of background noise in MR images biases the calculation of default WW/WL values since there has to be a trade-off between enhancing contrast of foreground/anatomy of interest vs suppressing background/ outside the anatomy of interest. A sub-optimal WW/WL would lead to poor visualization making human intervention necessary more often than needed to make the image readable. Also, as mentioned earlier, it is challenging to manually adjust WW/WL to achieve necessary contrast in anatomy and hiding the background pixels at the same time. All this may lead to visual fatigue for the viewer and consequently a delayed diagnosis.

This paper proposes a deep learning (DL) based method that improves the WW/WL for original as well as derived images (e.g., functional maps) by intelligently eliminating the unwanted background pixels. This also provides better control for user defined windowing in post processing as background pixels are already eliminated.

## 2. Literature review

Most of the literature for simple adaptive windowing focuses on suppressing background/ denoising via simple global thresholding (Felmlee et al., 1999). A slightly more advanced method uses thresholding followed by morphological operations to extract RoI (Belykh and Cornelius, 2007). Also, thresholding creates holes within anatomy leading to loss of information and hence sub-optimal windowing. There are other methods which uses segmentation as an alternative. For example, the adaptive windowing described in (Kaushik et al.) is specific to contrast enhancement of the structure of interest within the anatomy and uses a level-set based segmentation algorithm to extract the RoI. A review of traditional segmentation methods(Sharma and Aggarwal, 2010) shows that there is no universal algorithm for segmentation of every medical image and most of these methods require hand crafted tuning. The ideal segmentation technique for an image depends on the application and the body part that is imaged.

Alternatively, a deep learning (DL) based approach has the advantage that it is a generic method which performs equally well for all applications, anatomies and protocols. The existing methods using the same (Lai and Fang, 2005) are complex and intensive and needs user defined WW/WL values for the whole training set.

The proposed method overcomes the inherent limitations of the thresholding based segmentation and WW/WL calculation by using DL based background suppression and the same also provides additional benefit of enhanced control in manual adjustment of WW/WL which is not the case in (Lai and Fang, 2005) since it just computes the WW/WL through a neural network.

## 3. Methodology

The steps used to achieve the revised WW/WL are as follows.

1 Obtain background suppressed image.
2 Use the foreground pixels to compute the WW/WL.
3 Display the segmented image with the newly computed WW/WL

In step 1 above, our method uses U-net for implementing background suppression.It was trained with 2700 images covering brain and abdomen and the corresponding masks representing the anatomy. We propose to use background suppressed image with auto computed WW/WL for all the further reading and processing. This will ensure that not only the image (original/ post processed) is displayed with improved window settings but also provide better control on dynamically adjusting the WW/WL afterwards.

## 4. Results

The accuracy of auto-windowing computation depends on the accuracy of segmentation, our evaluation strategy was twofold- evaluating the accuracy of U-net for segmentation followed by image review from clinical application specialists.

1 U-net was used to perform segmentation on 1130 test images covering brain and abdomen and an average DICE score of 0.94 (min = 0.713; max = 0.982) was achieved.

2 The background suppressed images were presented with a better WW/WL than the original images(refer Figure 1)

3 It was also confirmed that the background suppressed image provided better control while dynamically adjusting WW/WL.

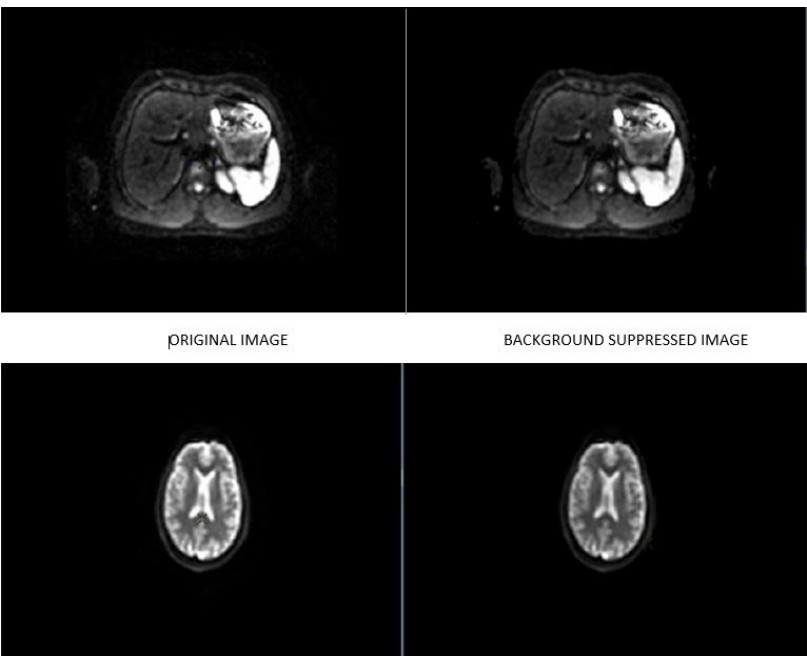

Figure 1: WW/WL before and after background suppression

## 5. Conclusion

The proposed solution addresses the problem of poor image quality and suboptimal WW/WL due to presence of background without compromising the anatomical details. The DL based segmentation demonstrates effective background suppression with high DICE scores. The WW/WL of both original as well as derived images can be significantly improved by pre-processing using proposed method.

## Acknowledgments

We would like to thank Ananthakrishna Madhyastha, MR Viz and Recon Manager and Venkatesh Seshachar, MR Software Engg Director from GE Healthcare for all the support and encouragement that helped make this paper.

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
