# OpenReview forum: "OPTIMAL WINDOWING OF MR IMAGES USING DEEP LEARNING: AN ENABLER FOR ENHANCED VISUALIZATION"
_MIDL.io/2019/Conference/Abstract — MIDL Abstract 2019_

### Official Review · AnonReviewer2 · 2019-04-30
**Simple but effective method for improved visualization, but lacks proper evaluation**

**Rating:** 2
**Confidence:** 1

**Review:**

This paper address the problem of finding automatically optimal values for Window Width (WW) and Window Level (WL) in MR images. Background and background noise will make it harder to find optimal values to visualize the object of interest.
The proposed solution is simple yet effective: a FCN perform a binary segmentation, separating foreground from background. Then, the predicted background is masked, and the usual WW/WL algorithm are performed on the remaining of the image. Due to the "simplicity" of the task (background detection), a single network could be used for the whole body.

Pros:
- Simple yet effective method
- Good results for the background segmentation

Cons:
- Lack proper evaluation of the final results, both quantitatively and qualitatively

Minor:
- The two cherrypicked examples are not that convincing, as it is difficult to see the noise/background in the first place.

---

### Official Review · AnonReviewer1 · 2019-05-01
**Background suppresion using a U-Net is applied before computing window width and window level**

**Rating:** 3
**Confidence:** 1

**Review:**

* The idea in this abstract is to prevent background pixels from affecting the WW and WL computations. This is achieved by using a supervised CNN to segment out the background and then compute the WW and WL using standard techniques.

* The authors claim that this leads to better visualization and can be applied to not only the original images but also derived images such as functional maps.

* As quantitative evaluation of visualizations is difficult, only qualitative results are shown. However, perhaps it would be better if more qualitative results are provided that better convey the problems without the background suppression.

---

### Decision · Program_Chairs · 2019-05-06
**Acceptance Decision**

Accept